# Benzodiazepine-Receptor Agonist Utilization in Outpatients with Anxiety Disorder: A Retrospective Study Based on Electronic Healthcare Data from a Large General Tertiary Hospital

**DOI:** 10.3390/healthcare11040554

**Published:** 2023-02-13

**Authors:** Denong Liu, Qingyu Zhang, Zhijia Zhao, Mengjia Chen, Yanbin Hou, Guanjun Wang, Haowei Shen, Huaqiang Zhu, Yunxin Ji, Liemin Ruan, Zhongze Lou

**Affiliations:** 1School of Medicine, Ningbo University, Ningbo 315211, China; 2Department of Psychosomatic Medicine, Zhejiang Regional Medical Center Ningbo First Hospital, Ningbo Hospital of Zhejiang University, Ningbo 315010, China; 3Department of Pharmacy, Ningbo Yinzhou No.2 Hospital, Ningbo 315199, China; 4Department of Pharmaceutical Engineering, Zhejiang Pharmaceutical University, Ningbo 315199, China; 5Zhejiang Key Laboratory of Precision Medicine for Atherosclerotic Diseases, Central Laboratory of the Medical Research Center, Zhejiang Regional Medical Center Ningbo First Hospital, Ningbo Hospital of Zhejiang University, Ningbo 315010, China

**Keywords:** anxiety, benzodiazepines, drug utilization, prescriptions, multiple drugs

## Abstract

Benzodiazepine-receptor agonists (BZRAs), including benzodiazepines (BZDs) and drugs related to BZDs (Z-drugs), are commonly used for anxiety, but often have side effects. We retrospectively investigated the utilization and prescription characteristics of BZRAs for patients with anxiety disorders in a large tertiary care general hospital between 2018 and 2021, based on electronic healthcare records. We also examined the pattern of simultaneous consumption of multiple BZRA drugs, and the diseases coexisting with anxiety that are associated with this. The numbers of patients and BZRA prescriptions increased over the 4 years. Moreover, 7195 prescriptions from 694 patients contained two or more BZRAs, of which 78.08% contained both BZDs and Z-drugs, 19.78% contained multiple BZDs, and 2.14% contained multiple Z-drugs. For anxiety patients with concomitant Alzheimer’s disease or Parkinson’s disease, and dyslipidemia, they were more likely to consume multiple BZRAs simultaneously, whereas patients with concomitant insomnia, depression, hypertension, diabetes, or tumors were less likely to consume multiple BZRAs (all *p* < 0.05). Furthermore, older patients who consume multiple BZRAs simultaneously may have higher probabilities of long-term drug use. Better interventions supporting standardized BZD utilization may be needed to minimize the side effects of inappropriate BZRA administration.

## 1. Introduction

Anxiety is common among all individuals, and appropriate anxiety can enhance working and learning efficiency, and help us better adapt to the rapidly developing society. Nevertheless, excessive or inappropriate anxiety can readily develop into anxiety disorders [1]. Anxiety disorders are one of the most common mental disorders, inflicting great physical and psychological suffering, severely reducing the quality of life of the sufferers [2,3], and imposing a tremendous economic burden on the global economy [4,5]. In a systematic review of epidemiological studies conducted in 44 countries, the prevalence of anxiety disorders globally was approximately 7.3%, which indicates that approximately 1 in 14 people worldwide is struggling with anxiety disorder at any given time, and approximately 1 in 9 may develop anxiety disorder within a given year [6]. Benzodiazepine-receptor agonists (BZRAs), including benzodiazepines (BZDs) and drugs related to BZDs (Z-drugs) are among the most commonly prescribed psychotropic drugs in clinical practice worldwide [7,8,9]. The pharmacological mechanism by which BZRAs exert sedative-hypnotic effects is mainly through binding to specific sites on the γ-aminobutyric acid (GABA) type A (GABAA) receptor, thereby potentiating the GABA’s inhibitory effect. According to medical guidelines and clinical practice, psychotherapy, or pharmacotherapy can effectively relieve the symptoms of patients with anxiety disorders. BZRAs are among the most commonly used groups of drugs for patients with anxiety disorders worldwide. Furthermore, they are also commonly used for sleep disturbance, phobic disorders, panic attacks, post-traumatic stress disorders, and epilepsy [10].

Most BZRAs are dispensed by primary healthcare centers, as many patients prefer to visit these places and obtain their medications for convenience or to save time [7]. Nevertheless, psychiatrists dispense a higher rate of prescriptions [11]. Many studies have reported considerably high consumption rates of BZRAs worldwide [12,13,14]. BZRA drugs have been clinically offered as the primary treatment for anxiety disorders for decades [15,16,17,18]. However, as a result of claims that BZRAs display a very high risk of withdrawal, rebound, overuse, and misuse [19,20,21,22,23], many studies have recommended selective serotonin reuptake inhibitors (SSRIs), or serotonin norepinephrine reuptake inhibitors (SNRIs) as the first choice medication for the treatment of anxiety disorders [24,25], an opinion that certainly gained grounds in clinical practice by the end of the 20th century. However, the new SSRIs and SNRIs have similar problems of withdrawal and rebound as BZDs, which had been reported in many studies [26,27].

Therefore, it is very difficult to completely replace BZRAs with SSRIs or SNRIs in the treatment of anxiety disorder. In addition, the literature provides evidence that BZRAs alone or in combination with other medications are an essential pharmacological approach for the treatment of anxiety disorders [28,29]. Despite these proposals, the use of BZRAs is indeed a worrying issue in the public health systems of many countries owing to their very high prevalence of prescriptions and the associated risks of long-term exposure [15,30]. Nevertheless, BZRAs increase the risk of falls, fractures, traffic accidents, detrimental cognitive effects, and Alzheimer’s disease (AD) [31,32,33,34,35]. Furthermore, other features, such as fear of tripping, falling, loneliness, social isolation, or reduction in social participation, may be adverse effects of treatment with BZRAs [36]. All these aspects also generate a huge economic burden; for example, the additional cost of BZRAs-related injury to falls in the European Union (EU) is estimated at 1.8 billion euros per year [37]. However, only 20–30% of BZRA prescriptions are appropriate in the clinic, according to previous studies [38,39]. Therefore, patients with anxiety must be prescribed appropriate BZRAs by clinicians for relief, cure, long-term physical and mental health development, and quality of life improvement, and for the rational allocation and utilization of medical resources.

This study aimed to determine the characteristics of BZRAs consumption in patients with anxiety in a large tertiary general hospital in China over a period of 4 years. Based on the results of the current study and previous studies, we offer recommendations to clinicians and patients to avoid the side effects of BZRAs.

## 2. Materials and Methods

The data used in this study were sourced from the electronic healthcare records of a large general tertiary hospital in Ningbo. The sample data may not be representative of all population levels in the region. Nevertheless, the general tertiary care hospital where this study was conducted is a general hospital in the region with the highest number of psychiatric outpatients. Furthermore, on the basis of our clinical observations and that of available studies reported, due to stigma and other reasons, many patients with psychological problems, such as anxiety, may prefer to seek assistance in the department of psychiatry of general hospitals instead of specialized psychiatric hospitals [40,41]. Patients who plan to obtain medication only on an outpatient basis and who do not plan to opt for inpatient treatment are more likely to choose a general hospital psychiatric department rather than a specialty psychiatric hospital [42]. Therefore, we believe that the data enrolled in this study can more effectively reflect the true status of BZRAs use among patients with anxiety disorder in this region. In general, when a patient first visits a hospital psychiatric clinic, a professional psychiatrist will conduct a comprehensive assessment of the patient through the patient’s medical history and main symptoms and relevant diagnostic questionnaires, and rule out organic brain disease by imaging tests, after which a diagnosis of anxiety disorder is made based on the examination results and relevant information already obtained, and a similar procedure is often used for the diagnosis of somatic disorders. Doctors usually follow the diagnostic criteria of the DSM-5 diagnostic manual when making diagnoses, after which a prescription is prescribed based on the patient’s main symptoms and diagnosis, through which the patient can obtain the relevant medication at the hospital pharmacy. At each subsequent follow-up visit, the doctor will make diagnoses through relevant tests while assess the effectiveness of the previous medication to determine whether to continue the previous medication or change the treatment medication. All prescription data will be stored in the electronic prescription record system.

Prescription records were obtained directly from the electronic medical record system database. This population-based database is currently available and provides information on prescribed drugs using the Anatomical and Therapeutic Classification of Medicines (ATC) [43]. Patients usually have an interview with a professional clinician when they visit and are assessed using a professional anxiety questionnaire, at least at the first visit. In the electronic healthcare record system of the hospital, diagnoses of diseases, date of prescription, physician identification number, and patient identification number were recorded. The ATC system is an internationally accepted classification system that divides all medical products into 14 anatomical main groups, each with two therapeutic subgroups and two chemical subgroups [43]. BZRAs offered by our hospital include BZDs(clonazepam[N03AE01], diazepam[N05BA01], lorazepam[N05BA06], alprazolam[N05BA12], and eszopiclone[N05CD04]), and Z-drugs (zopiclone[N05CF01] and zolpidem[N05CF02]). Only BZRA prescriptions for outpatients with the diagnosis of anxiety disorder were enrolled, and we assumed that the patients took their medications in accordance with the prescriptions. Similar to the protocols followed in France and Croatia, in the hospital where this study was implemented, prescriptions including BZRA drugs are dispensed for a maximum of approximately 3–4 weeks, regardless of the actual BZRA drug included in the prescription [44,45].

Many patients had coexisting physical or psychiatric disorders, in addition to anxiety disorder. Therefore, these influential factors, including other somatic or psychiatric disorders, age, and sex, were analyzed to assess their contribution to the types of drugs in BZRA prescriptions that were prescribed to the patients with anxiety disorder in the clinical outpatient service. Therefore, we analyzed a range of common somatic disorders and psychiatric disorders to assess their impact on the types of prescriptions obtained by patients with anxiety disorders. The prescriptions associated with other concomitant disorders, which were large in number, but were not presented in detail by us, were not analyzed because the number of prescriptions associated with each disorder was small.

Independent samples t-tests and chi-squared tests were used to compare the differences. Univariate analysis was performed using Student’s t-test for continuous variables, depending on the distribution of the data, and the chi-squared test for categorical variables. Multivariate logistic regression analyses were used to assess the association of the variables of other physical or psychiatric disorders that coexist with anxiety in patients with the number of BZRA drug types of prescriptions dispensed by the clinicians. Variables with *p* values < 0.05 in the univariate analyses were considered potential factors for inclusion in the multivariate logistic model, and all of these variables were entered into the final model. Statistical analyses were performed using SPSS (IBM SPSS Statistics for Windows, Version 25.0. Armonk, NY, USA: IBM Corp.). A two-sided *p* < 0.05 was considered statistically significant.

## 3. Results

The cohort comprised BZRAs consumers at a large tertiary general hospital during the period of 2018–2021. This study encompassed 7836 patients with anxiety disorders who generated 42,842 records of BZRA prescriptions dispensed for treatment, which included all possible BZDs and Z-drug consumption in patients with anxiety disorder. Table 1 shows the overall characteristics of the study population. There were 4941 women and 2895 men who had consumed BZRAs at least once, accounting for 63.1% and 36.9% of the total patient population, respectively. In addition, a large proportion of the enrolled patients were in the 25–64 years old age group, accounting for approximately 73.6% of the total patient population. The number of prescriptions consisting of BZRA drugs were 28,039 among female patients and 14,803 among male patients, accounting for 65.4% and 34.6% of the total number of prescriptions, respectively. Among all prescriptions, 35,647 (83.2%) contained only one BZRA drug, 6546 (15.3%) contained two, and 649 (1.5%) contained three or more. Each prescription of BZRAs dispensed by a clinician in outpatient services was accompanied by a relatively accurate clinical diagnosis. Since patients with anxiety disorder at the first visit, the second visit, the last visit, etc., may have different concomitant disease diagnoses; therefore, we analyzed the data from the last visit of all patients, there are probably 2663 (34.0%) patients with anxiety disorder in this study who do not have other coexisting disorders. Of the clinical diagnoses of co-occurring anxiety disorder in relation to BZRA prescriptions, insomnia is the most common, accounting for approximately 38.2%. Moreover, other coexisting diagnoses of anxiety disorders, including psychiatric disorders, such as depression, Parkinson’s disease (PD), Alzheimer’s disease (AD), and bipolar disorder, and common chronic physical diseases, such as hypertension (HTA), dyslipidemia, diabetes, coronary artery disease (CHD), and tumors, have also been observed. In addition, approximately 4376 (55.9%) patients with anxiety disorders received only one BZRA prescription during the entire study period.

The total number of prescriptions increased over the 4 years of the study sampling and had the largest annual growth rate in 2019, at 66.3% (Figure 1A). Moreover, the number of prescriptions for women has been on an upward trend since 2018, with the largest annual growth rate in 2019, at 70.6%. The number of prescriptions for male patients with anxiety had the largest annual growth rate in 2019, at 58.8%, while there was also a decline in 2021, with a decline rate of 4.1% (Figure 1 A). The total number of patients with anxiety disorder has been on an upward trend since 2018, and has the largest annual growth rate of 47.8% in 2021. The largest growth rates in the number of female patients and of male patients were in 2021 and 2019, at 49.1% and 48.7%, respectively (Figure 1B).

Outpatient BZRAs use according to age and sex from 2018 to 2021 is shown in Table 2. In relation to sex, women with anxiety disorders generated significantly higher prescriptions for BZRAs than men in each of the four years from 2018 to 2021 (*p* < 0.05; 63.29% BZRA prescriptions from women in 2018, 64.95% BZRA prescriptions from women in 2019, 65.62% BZRA prescriptions from women in 2020, and 66.94% BZRA prescriptions from women in 2021) (Table 2). Older age groups generated a higher number of prescriptions than that of younger age groups, with the highest percentage found in the age group of 45–64 (Table 2). The average number of BZRA prescriptions utilized per patient in 1 year was the highest in the oldest age group (>64 years), with averages of 5.40, 6.78, 6.41, and 4.12 prescriptions per patient per year, for the years 2018–2021, respectively (Table 2). Furthermore, the annual average number of prescriptions in all age groups increased in 2019 compared to that in 2018, except for the 16–24 age group (Table 2). In 2020 and 2021, the annual average number of prescriptions for most age groups remained consistently lower than that in 2019 and 2020, respectively (Table 2). There was no significant difference in the number of annual average prescriptions between men and women during the 4 years; however, we found an apparent decrease in the number of annual average prescriptions per patient in 2021 compared to that in 2020 for total, men, and women, with decline rates of 32.40%, 34.00%, and 31.81%, respectively.

The number of prescriptions containing only one BZRA drug increased by 4024 over the 4 years, with the annual largest growth rate observed in 2019 at 67.7%, and the number of prescriptions in 2019 increased by 3972 compared to that in 2018 (Figure 2A). The number of patients associated with BZRAs prescriptions containing only one BZRA drug increased by 1768 over the 4 years, with women accounting for 69.0% of the increasing number of patients with anxiety disorder (Figure 2B). The number of prescriptions containing multiple BZRA drugs increased by 1420 over the 4 years, with the annual largest growth rate observed in 2019 at 58.1%, and an increase of 567 prescriptions compared to 2018 year (Figure 2A). Furthermore, the number of patients with anxiety disorder associated with increased prescriptions containing multiple BZRA drugs increased by 211 over the 4 years, with women accounting for 78.7% of the increasing number (Figure 2B). In addition, the annual mean number of prescriptions per patient using prescriptions containing multiple BZRAs was higher than that of patients receiving prescriptions containing only one BZRA drug in any year (*p* < 0.001, Figure 2C). The annual average number of prescriptions decreased dramatically over the 4 year period for both patients who received prescriptions, including only one BZRA drug and those who received multiple BZRA drugs. Moreover, the annual average number of prescriptions decreased at a greater rate of 30.7% for patients given prescriptions containing two drugs than the 27.1% decrease observed in those who obtained prescriptions containing only one BZRA drug over the 4 years (Figure 2C).

To explore the sex-specific differences in the types of BZRA drugs administered to patients with anxiety disorder, we separately screened prescriptions for anxiety disorder that included only one BZRAs drug. Furthermore, we excluded prescriptions obtained by patients with any comorbid physical or mental disorders in order to exclude their possible impact on the type of BZRA drugs prescribed. A total of 15,658 prescription samples for anxiety were screened from patients with anxiety who did not have any comorbid physical or psychiatric disorders. Among BZDs, lorazepam had the highest rate of use in the total sample, followed by estazolam. Among the Z-drugs, zolpidem had a higher rate of use in the total sample than that of zopiclone (Table 3). In addition, clonazepam, lorazepam, and zopiclone were more likely to be consumed by men, whereas zolpidem was more likely to be consumed by women (Table 3).

The number of prescriptions containing both BZDs and Z-drugs was 5618, prescriptions containing only BZD drugs numbered 1423, and prescriptions containing only two types of Z-drugs numbered 154, accounting for 78.08%, 19.78%, and 2.14% of the total number of prescriptions containing two or more types of BZRA drugs, respectively (Figure 3).

The number of prescriptions containing only one BZRA drug and the associated anxiety patients both increased with age increasing and that the maximum number of prescriptions and associated patients were observed in the 45–64 year old group. The average number of prescriptions, including those of only one BZRA drug per patient, increased progressively with increasing age, especially in the >64 year old age group, which had the largest average number of prescriptions per patient regardless of the total or of sex (Table 4). A similar result was observed in patients who received prescriptions containing multiple BZRAs (Table 5). Moreover, regardless of age group, the latter was higher than the former in terms of the number of average prescriptions per patient, regardless of age group (*p* = 0.022, Table 4 and Table 5).

Ten somatic and psychiatric disorder variables commonly coexisting with anxiety disorders were determined, including insomnia, depression, PD, AD, bipolar disorder, HTA, dyslipidemia, diabetes, CHD, and tumors. Univariate analysis showed that age, sex, insomnia, depression, PD, AD, HTA, dyslipidemia, diabetes, and tumors were potential factors related to the types of BZRAs prescribed to patients with anxiety disorder, whereas bipolar disorder and CHD were not. Sex, age, and the eight variables of somatic or psychiatric diseases that provided significance in the univariate analyses were chosen as independent variables, which were included in the multivariate logistic regression analysis model. Multivariate logistic regression analysis demonstrated that older patients with anxiety disorders were more likely to obtain multiple BZRAs simultaneously (OR = 1.018, 95% CI: 1.016–1.019). Although more female patients with anxiety disorders had been repeatedly offered multiple BZRAs concurrently, based on the results of our analysis, female patients with anxiety disorders did not tend to be prescribed multiple medications simultaneously more than male patients with anxiety disorders were (OR = 0.882, 95% CI: 0.835–0.932). Furthermore, patients with anxiety who suffered from one of these somatic or mental disorders, such as PD (OR = 27.161, 95% CI: 10.134–72.796), AD (OR = 15.414, 95% CI: 4.893–48.556), or dyslipidemia (OR = 2.566, 95% CI: 2.002–3.289), were more likely prescribed two or more BZRAs simultaneously. The maximum contribution was from PD, followed by AD. In addition, multivariate logistic regression analysis also demonstrated that patients with anxiety disorder who simultaneously suffered from insomnia (OR = 0.617, 95% CI: 0.586–0.649), depression (OR = 0.614, 95% CI: 0.562–0.672), HTA (OR = 0.783, 95% CI: 0.693–0.885), diabetes (OR = 0.638, 95% CI: 0.521–0.780), or tumors (OR = 0.445, 95% CI: 0.303–0.652) were more likely to use only one BZRA drug (Table 6). 

## 4. Discussion

This study revealed the prescription trends of BZRAs using the hospital dispensary database in a large-scale general tertiary hospital in Ningbo, China, over a period of 4 years from 2018 to 2021. The patients with anxiety disorder consuming BZRAs at least once, who are covered by this study; the number of female patients and the dispensed prescription records generated by them were approximately twice as high as those of male patients with anxiety disorder. Similar investigations have also been reported in the United States, Canada, and Europe on BZRA drug usage [46,47,48]. With regard to drug indications, clinical outpatient clinicians prescribed BZRAs for the diagnosis of anxiety disorders in this study. Since each diagnosis is recorded concurrently with the dispensation of prescriptions by the clinical outpatient physicians, it is commonly known that some patients with anxiety disorders may have other comorbidities. The clinical comorbidity diagnoses may not be constant each time the same person receives a prescription from a physician; thus, we analyzed the related concomitant diagnoses according to the total prescriptions. The most common psychiatric disorder diagnosis coexisting with anxiety disorders was insomnia, followed by depression. The most common somatic disorder diagnosis coexisting with anxiety disorders was hypertension, followed by dyslipidemia. Many studies have demonstrated that anxiety may be a hazardous factor in sleep disturbance [49,50]. Individuals with anxiety disorders are susceptible to dysfunctional sleep arousal, which may lead to a state of persistent sleep disruption [51,52]. The correlation between anxiety, hypertension, and dyslipidemia has also been reported in many studies [53,54,55].

The total number of prescriptions exhibited an upward trend from 2018 to 2020; however, compared to that in 2020, the number of prescriptions showed a slight decline in 2021, which is primarily due to the decrease in the number of prescription records generated by male patients with anxiety compared to those in 2020. In addition, we found a notably low growth rate in the number of people prescribed BZRAs in 2020 compared to those in 2019 and 2021. Although we did not explore the reasons in-depth, we speculate that some patients with anxiety may be reluctant to visit hospitals distant from their homes because of the epidemic outbreak of the coronavirus disease 2019 in late 2019, and due to some strict quarantine measures implemented in China thereafter [56]. Thus, these patients may prefer to go to community hospitals or other hospitals closer to their homes to obtain relevant BZRA drugs. In addition, we found that the annual average number of BZRA prescriptions for patients with anxiety disorder increased consistently with increasing age, and that the highest mean annual number of prescriptions was observed in the >65 year old age group. 

Although we did not explore the duration of drug intake by the patients in this study, according to our inquiry, BZRAs were prescribed to patients for approximately 3–4 weeks at a time in the hospital; thus, the annual average number of prescriptions could indirectly reflect the duration of BZRA medication use. Therefore, our results also indirectly suggest that elderly patients with anxiety who consume multiple BZRA drugs prefer to have a longer duration of BZRA administration. This phenomenon of long-term or overdose of BZRAs in the elderly has also been reported in many similar studies [7,57,58]. The side effects of long-term BZRAs utilization or over-usage in the elderly are considerable, and it can substantially increase the risk of dementia, traffic accidents, falls, and fractures [59,60]. Previous studies suggest that caution should be exercised when BZRA prescriptions are dispensed to older patients [61]. In addition, other aspects, such as fear and worry about falls, loneliness, social isolation, or recognition of damage, can be side effects of BZRA treatment, and this may ultimately lead to increased negative moods and decreased social participation [36], which may result in a higher risk of negative emotions, an increased risk of taking BZDs, and subsequently an increased probability of BZRA side effects [62]. A vicious circle eventually forms between BZRAs treatment and its side effects. According to EU statistics, the additional annual costs associated with the use of BZRAs and their related side effects, including falls, fractures, and so on, are approximately 1.8 billion euros in Europe [37]. Nevertheless, no international consensus has been reached on the duration of BZRAs use. Although some treatment guidelines recommend that BZRAs should not be used for more than 6 weeks (including tapering before withdrawal) for anxiety treatment [63,64,65], and most physicians are profoundly aware of the risks associated with the long-term use of BZRAs in older individuals in the guidelines, many still do not consider it a serious clinical threat, and many do not feel the necessity or readiness to address this concern with the patients [58]. Encouragingly, in 2021, the annual average number of prescriptions for patients with anxiety disorder. Similar studies have recently reported a trend of decreasing prescription rates for BZRAs [66]. We believe that the decline in annual average prescriptions in 2021 may be attributed to the presence of more patients with anxiety disorders in 2021 for the reason of COVID-19 epidemic, which has also been reported in some of the recent literature [67,68]. In addition, this indirectly indicates that despite the many problems encountered in the clinical application of BZRA drugs, including their poor clinical efficacy, the development of drug resistance, and adverse effects after discontinuation, clinicians may have been making continuous efforts to align with the guidelines for reducing the long-term use of BZRAs drugs for the maximum well-being of patients. 

Furthermore, many guidelines suggest that regimens involving multiple BZRAs should be simplified and converted to single-BZRA prescriptions internationally [69,70,71]. Concurrent consumption of multiple BZRAs can result in exaggerated pharmacological effects, unintended overdosing, and other potential adverse reactions. While the concurrent use of multiple BZRAs may be unintentional, it often arises when patients have multiple prescribers and/or obtain a prescription that contains multiple BZRAs drugs. Therefore, to minimize the risk of adverse effects and to set the stage for an attempt to wean older adults off BZRAs, multiple-BZRA regimens must be simplified to single-BZRA regimens. However, in our study, approximately 7.5% of patients consumed at least one prescription containing two or more type BZRAs during the 4 years, and a total of 7195 prescription records were generated, with 78.08% of these prescriptions containing both BZDs and Z-drugs, 19.78% containing multiple BZDs, and 2.14% containing multiple Z-drugs. Furthermore, patients who consumed multiple BZRAs simultaneously had higher annual average prescriptions than those of patient who consumed only one BZRA, and the annual average number of prescriptions increased significantly with increasing age. This also indirectly indicates that older patients who consumed multiple BZRA medications simultaneously are more inclined to take those medications for a longer time.

Furthermore, patients who experience anxiety in conjunction with PD, AD, or dyslipidemia were usually more likely to consume multiple BZRAs, whereas patients with anxiety disorder in conjunction with depression, hypertension, diabetes, or tumors usually tended to consume only single BZRAs. Although anxiety are very common in patients with PD and AD, they have been grossly overlooked [72,73]. Furthermore, anxiety in patients with PD and AD are related to severely diminished quality of life [74,75]. The treatment of anxiety in patients with PD or AD has been unsatisfactory; therefore, this may be one of the reasons for their overuse or abuse of BZRA drugs [73]. Since the time point of disease onset and medication use were not analyzed, we could not exclude the possibility that the patient took multiple BZRA drugs early in life, which have ultimately increased the occurrence of PD or AD. However, numerous studies have shown that BZRAs are important risk factors and predictors of the development of AD and PD in elderly patients [76,77,78,79]. In addition, when anxiety coexisted with depression, sleep disorders, and other chronic physical disorders, multiple BZRAs did not tend to be consumed simultaneously by the patients. This may be due to the possibility that when anxiety disorders are comorbid with these psychiatric or somatic disorders, patients may also receive other drug treatments, and other therapeutic approaches, such as psychotherapy, cognitive-behavioral therapy, and physical therapy, which are often used for disease intervention. In addition, when anxiety patients have comorbid insomnia or other chronic physical illnesses, they may be more inclined to have a healthier lifestyle, which includes emphasis on physical activity, regular routine, healthy diet, and good lifestyle habits, which may indirectly reduce drug dependence, overuse, or abuse.

We also analyzed the prescriptions generated by patients with anxiety who consumed only one BZRA drug and found that lorazepam had the highest percentage of consumption, while zolpidem accounted for a higher percentage of Z-drug consumption. Among the BZDs, lorazepam and clonazepam were more frequently used by males. Among the Z-drugs, zolpidem was more frequently used by women, while zopiclone was more frequently used by men. This sex difference in BZRA drug consumption is roughly consistent with that reported by a previous study, wherein BZDs were more likely to be used by men, while Z-drugs were mostly consumed by women [80]. Unfortunately, no data from the literature have evaluated sex distribution among BZDs and Z-drugs in detail; thus, the reasons for this phenomenon are still not clearly explained. 

Based on the present findings, some recommendations can be made for both research and clinical practice. First, increased education of clinicians and pharmacists, and increasing patients’ knowledge about BZRAs drugs should be implemented so that inappropriate use of BZRAs can be predicted and stopped at the time of initiation, which may be a more efficient way to optimize the use of BZRAs than looking for alternative solutions when inappropriate use is detected. Second, clinicians should strengthen comprehensive and individualized assessments during treatment, fully establish doctor-patient interactions, identify patients who may be susceptible to dependence or addiction at an early stage, and take measures to make optimal decisions for different patients, diseases, and disease courses. Third, non-pharmacologic treatments should be recommended first to relieve anxiety in patients, such as cognitive-behavioral therapy, physical therapy, and exercise therapy. If medication is an essential option, try to avoid prescribing multiple BZRA drugs and the long-term use of BZRA drugs. Fourth, enhance the utilization of prescription data and adopt information technology to improve scientific supervision before and after prescription. Finally, policy measures, including prescription monitoring and formulary restrictions, should also be taken to optimize the use of BZRAs and ultimately avoid their inappropriate use and the side effects caused by inappropriate BZRA drug use.

This study had several limitations. First, the prescription data identified purchased medicines rather than medication use. Second, while some information on indication is available through the electronic medical record system, we could not ascertain whether any prescriptions were truly inappropriate. There may be exceptional circumstances in which long-term BZRAs treatment or experimental treatment with drug co-administration with follow-up and supervision may be acceptable for some patients. Third, since prescriptions for BZRAs are dispensed by clinicians based on the patient’s symptoms, financial status, and the clinicians’ clinical experience, we did not collect this information; thus, the analysis of the results may be biased.

## 5. Conclusions

This study described the trends regarding BZRAs use in patients with anxiety disorders at a large general tertiary hospital over a 4 year period (2018–2021). The use of BZRAs remains a concern in terms of safety; however, many patients continue to take BZRAs to relieve their anxiety symptoms. During this 4 year period, both the number of patients with anxiety using BZRAs medications and the number of BZRA prescriptions generated by them showed increases. Moreover, the number of prescriptions for BZRAs was the highest among elderly female patients with anxiety. Moreover, elderly patients with anxiety disorder tended to have a higher average number of prescriptions per year. Nonetheless, there was a decline in the annual average number of prescriptions per patient over the 4 year period, among both male and female patients with anxiety disorders. Meanwhile, this study exposed a concern of the group of patients who co-consumed multiple BZRAs drugs. This group tended to have a much higher annual average number of BZRA prescriptions, especially among the elderly. In addition, patients with AD or PD who simultaneously suffered from anxiety disorders are much more likely to be co-consuming multiple BZRA drugs. Since many other non-pharmacological alternatives, such as psychotherapy and cognitive-behavioral therapy, are available, efforts should be made to educate physicians and patients about these alternatives to avoid side effects related to long-term use, abuse, or overdose of BZRAs as much as possible.

## Figures and Tables

**Figure 1 healthcare-11-00554-f001:**
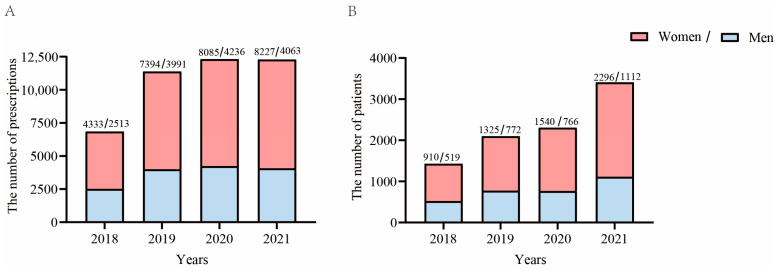
Trends of number of BZRA prescriptions and related patients prescribed BZRAs by sex from 2018 to 2021. (**A**) Trends of the number of prescriptions of BZRA drugs dispensed for patients with anxiety disorder from 2018 to 2021. (**B**) Trends of the number of patients with anxiety disorder related to prescriptions of BZRA drugs from 2018 to 2021.

**Figure 2 healthcare-11-00554-f002:**
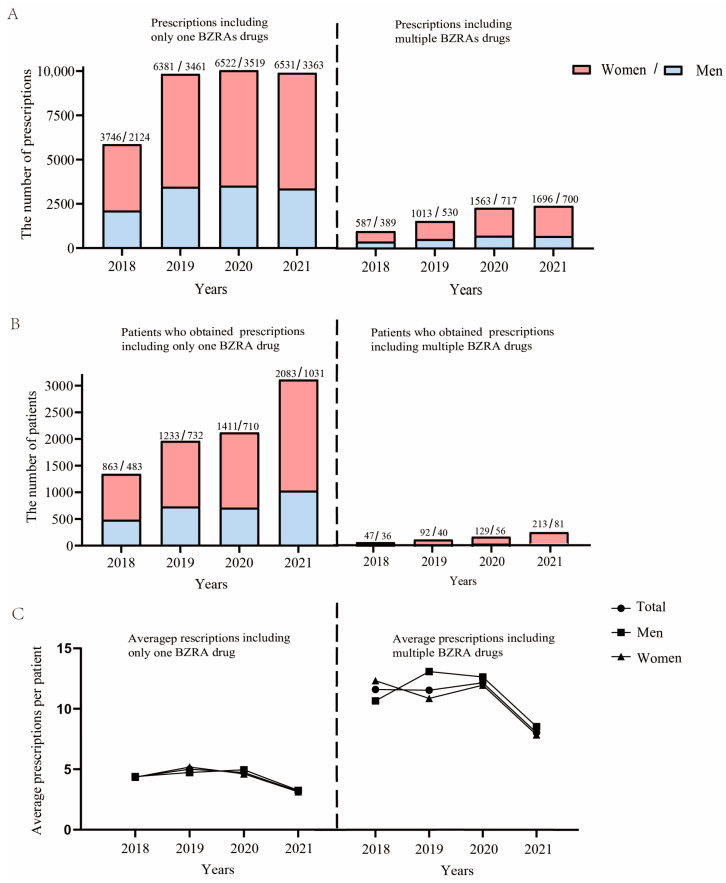
Trends of the number of prescriptions of only one and two or more types of BZRA drugs and the associated number of patients by sex from 2018 to 2021. (**A**) Trends in the number of prescriptions containing only one BZRA drug or two or more BZRA drugs from 2018 to 2021. (**B**) Trends in the number of outpatients who were dispensed with prescriptions containing only one BZRA drug or containing two or more BZRA drugs from 2018 to 2021. (**C**) Trends in the average annual number of prescriptions per patient dispensed containing only one BZRA drug or containing two or more BZRA drugs from 2018 to 2021.

**Figure 3 healthcare-11-00554-f003:**
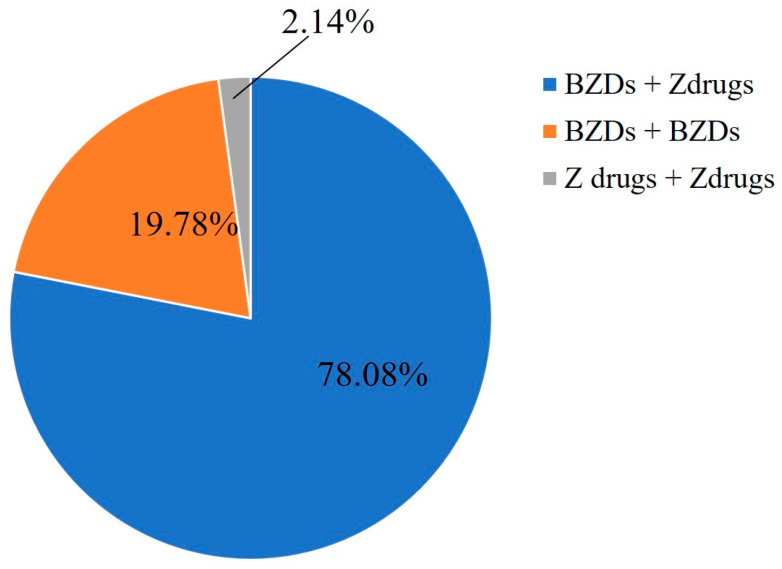
Respective composition ratios of prescriptions containing multiple BZRA drugs, which contain both BZD drugs and Z-drugs, only multiple BZDs, and only multiple Z-drugs simultaneously.

**Table 1 healthcare-11-00554-t001:** Study population and BZRAs prescription characteristics between 2018 and 2021.

Classification	Population n (%)	Prescription n (%)
Sex		
Men	2895 (36.9%)	14,803 (34.6%)
Women	4941 (63.1%)	28,039 (65.4%)
Age		
<16	86 (1.1%)	213 (0.5%)
16–24	577 (7.4%)	1905 (4.5%)
25–44	2625 (33.5%)	11,882 (27.7%)
45–64	3144 (40.1%)	18,768 (43.8%)
*>*64	1404 (17.9%)	10,074 (23.5%)
Other accompanying diseases		
Sleep disorders	3611 (46.1%)	16,404 (38.2%)
Depression	870 (11.1%)	3218 (7.5%)
PD	709 (9.1%)	445 (1.0%)
AD	54 (0.7%)	156 (0.4%)
Bipolar disorder	9 (0.1%)	45 (0.1%)
HTA	321 (4.1%)	1782 (4.2%)
Dyslipidemia	129 (1.7%)	803 (1.9%)
Diabetes	78 (1.0%)	538 (1.3%)
CHD	75 (1.0%)	220 (0.5%)
Tumors	20 (0.3%)	122 (0.3%)
Other	2573 (32.8%)	20,565 (48.0%)
No	2663 (34.0%)	18,230 (42.6%)
Types of BZRAs on prescription		
1	7530 (96.1%)	35,647 (83.2%)
2	806 (10.3%)	6546 (15.3%)
*≥*3	78 (1.0%)	649 (1.5%)
The number of prescriptions obtained by patients during the study period		
1	4376 (55.9%)	4376 (10.2%)
*≥*2	3460 (44.1%)	38,466 (89.8%)

Abbreviations: n, number; PD, Parkinson’s disease; AD, Alzheimer’s disease; HTA, hypertension; CHD, coronary atherosclerotic heart disease.

**Table 2 healthcare-11-00554-t002:** Prevalence of BZRAs utilization stratified according to age and sex from 2018 to 2021, expressed as the total number of BZRA prescriptions per patient and in amounts.

Age	2018				2019				2020				2021			
	Men	Women	Total	Average of Total Number of BZRA Prescriptions	Men	Women	Total	Average of Total Number of BZR Prescriptions	Men	Women	Total	Average of Total Number of BZRA Prescriptions	Men	Women	Total	Average of Total Number of BZRA Prescriptions
<16 (%)	8 (0.32)	11 (0.25)	19 (0.28)	1.46	11 (0.28)	31 (0.42)	42 (0.37)	3.50	15 (0.35)	61 (0.75)	76 (0.62)	3.30	12 (0.30)	64 (0.78)	76 (0.62)	1.69
16–24 (%)	84 (3.34)	149 (3.44)	233 (3.40)	3.24	157 (3.93)	291 (3.94)	448 (3.94)	3.00	199 (4.70)	287 (3.55)	486 (3.94)	2.98	271 (6.67)	467 (5.68)	738 (6.00)	2.19
25–44 (%)	758 (30.16)	1173 (27.07)	1931 (28.21)	4.44	1251 (31.35)	1920 (25.97)	3171 (27.85)	4.93	1272 (30.03)	2080 (25.73)	3352 (27.21)	4.73	1175 (28.92)	2253 (27.39)	3428 (27.89)	3.37
45–64 (%)	998 (39.71)	2088 (48.19)	3086 (45.08)	5.00	1507 (37.76)	3410 (46.12)	4917 (43.19)	5.59	1650 (38.95)	3916 (48.44)	5566 (45.17)	5.74	1519 (37.39)	3680 (44.73)	5199 (42.30)	3.94
≥65 (%)	665 (26.46)	912 (21.05)	1577 (23.04)	5.40	1065 (26.69)	1742 (23.56)	2807 (24.66)	6.78	1100 (25.97)	1741 (21.53)	2841 (23.06)	6.41	1086 (26.73)	1763 (21.43)	2849 (23.18)	4.12
Total number of BZRAs Prescriptions %	2513 36.71	4333 63.29	6846	4.79	3991 35.05	7394 64.95	11,385	5.43	4236 34.38	8085 65.62	12,321	5.34	4063 33.06	8227 66.94	12,290	3.61
Average of prescriptions by sex	4.84	4.76			5.17	5.58			5.53	5.25			3.65	3.58		

**Table 3 healthcare-11-00554-t003:** Propensity to be prescribed BZRAs for anxiety patients taking only one BZRA drug by sex.

				Sex		
Anxiety				Male	Female	Statistics
ATC Code	BZRAs	N	Total%	N (%)	N (%)	*p*
BZDs						
N03AE01	Clonazepam	1702	10.87	670 (12.10)	1032 (10.20)	<0.001
N05BA01	Diazepam	104	0.66	43 (0.78)	61 (0.60)	0.239
N05BA06	Lorazepam	6009	38.38	2218 (40.06)	3791 (37.46)	0.001
N05BA12	Alprazolam	1315	8.40	458 (8.27)	857 (8.47)	0.695
N05CD04	Estazolam	1898	12.12	633 (11.43)	1265 (12.50)	0.054
Z-drugs						
N05CF01	Zopiclone	1400	8.94	540 (9.75)	860 (8.50)	0.009
N05CF02	Zolpidem	3230	20.63	975 (17.61)	2255 (22.28)	<0.001

**Table 4 healthcare-11-00554-t004:** Number of prescriptions with only one BZRA drug and the number of relevant patients, and the average number of prescriptions by age and sex.

	<16	16–24	24–44	45–64	>64	Total
Men						
Prescriptions	43	654	3956	4684	3130	12,467
Individuals	23	214	943	956	581	2717
Average prescription	1.87	3.06	4.20	4.90	5.38	4.59
Women						
Prescriptions	164	1104	6347	10,536	5029	23,180
Individuals	61	338	1529	1894	710	4532
Average prescription	2.69	3.27	4.15	5.56	7.08	5.11
All						
Prescriptions	207	1758	10,303	15,220	8159	35,647
Individuals	84	552	2472	2850	1291	7249
Average prescription	2.46	3.18	4.17	5.34	6.32	4.92

**Table 5 healthcare-11-00554-t005:** Number of prescriptions with two or more BZRA drugs, and the number of relevant patients, and average number of prescriptions by age and sex.

	<16	16–24	24–44	45–64	>64	Total
Men						
Prescriptions	3	57	500	990	786	2336
Individuals	1	10	51	79	37	178
Average prescription	3.0	5.7	9.8	12.53	21.2	13.1
Women						
Prescriptions	3	90	1079	2558	1129	4859
Individuals	1	15	102	215	76	409
Average prescription	3.0	6.0	10.6	11.9	14.9	11.9
All						
Prescriptions	6	147	1579	3548	1915	7195
Individuals	2	25	153	294	113	587
Average prescription	3.0	5.9	10.3	12.1	17.0	12.3

**Table 6 healthcare-11-00554-t006:** Binary logistic regression analysis identifying the effect of coexisting diseases to the acquisition of prescriptions containing two or more BZRA drugs in patients suffering from anxiety disorders.

	Univariate Analysis		Multivariate Analysis	
Variables	OR (95% CI)	*p* Value	OR (95 %CI)	*p* Value
Age	1.016 (1.014–1.017)	<0.001	1.018 (1.016–1.019)	<0.001
Men	0.894 (0.847–0.943)	<0.001	0.882 (0.835–0.932)	<0.001
Insomnia	0.612 (0.581–0.644)	<0.001	0.617 (0.586–0.649)	<0.001
Depression	0.676 (0.620–0.738)	0.004	0.614 (0.562–0.672)	<0.001
PD	22.519 (8.412–60.286)	<0.001	27.161 (10.134–72.796)	<0.001
AD	10.334 (3.295–32.406)	<0.001	15.414 (4.893–48.556)	<0.001
Bipolar disorder	1.312 (0.556–3.101)	0.536	——	——
HTA	0.709 (0.632–0.796)	<0.001	0.783 (0.693–0.885)	<0.001
Dyslipidemia	2.071 (1.624–2.642)	<0.001	2.566 (2.002–3.289)	<0.001
Diabetes	0.558 (0.460–0.677)	<0.001	0.638 (0.521–0.780)	<0.001
CHD	1.067 (0.743–1.533)	0.725	——	——
Tumors	1.383 (0.263–0.557)	<0.001	0.445 (0.303–0.652)	<0.001

Abbreviations: n, number; PD, Parkinson’s disease; AD, Alzheimer’s disease; HTA, hypertension; CHD, coronary atherosclerotic heart disease.

## Data Availability

The datasets analyzed during the current study are not publicly available because hospital data were used, but are available from the corresponding author upon reasonable request.

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
