# Peer review of "Benzodiazepine-Receptor Agonist Utilization in Outpatients with Anxiety Disorder: A Retrospective Study Based on Electronic Healthcare Data from a Large General Tertiary Hospital"

_healthcare, 2023, doi:10.3390/healthcare11040554_

Round 1

Reviewer 1 Report

Summary

The authors have studied use of Benzodiazepine-reseptor agonists (BZRAs) – with focus on simultaneous use of different types of BZRAs – by analysing electronic health care records 2018-2021 from a large tertiary hospital in Ningbo, China. They found that 17% of the prescriptions contained two or more BZRAs, and that patients with Alzheimer’s and Parkinson’s disease were at particularly high risk for being prescribed multiple BZRAs (odds-ratio > 10). I think the authors could give some more information about the methods and the study setting, and about how they have handled the dependence between prescriptions to the same patients in the analyses.

General comments

Setting: I think it would be informative for international readers if the authors gave a description of the Chinese practice. They state that prescription records were obtained directly from the electronic medical record system database of a hospital in Ningo. Do the patients in the study get all their successive prescriptions from this hospital (except for the covid 19 period mentioned on page 12)? No follow-up by general practitioners? Do they get all prescriptions from the same physician in the hospital? No chance for ‘doctor-shopping’? Can they have only one dispensing per prescription? Can simultaneous use of several BZRA types occur even though each prescription only is for one type?

Study population/study period: They state that the number of patients is 9,240 (page 4, line 140). However, this equals the sum of yearly number of patients in Figure 1 B. Does that mean that no patients received prescriptions of BZRAs in different calendar years, or does it mean that the number of unique patients is less than 9,240? Is the study period from January 1, 2018 to December 31, 2021?

Accompanying diseases: The authors give the number of prescriptions per disease in table 1, but not the number of patients with the diseases. I think that should be included. The ‘other’ category is quite big. I assume that it contains disease categories that are bigger than most of the ones included in table 1. The authors could motivate the selection of accompanying diseases in the Materials and Methods section. And what was the percentage of prescriptions/persons that had no accompanying disease?

Methods: Have the authors taken into account that prescriptions are not independent entities, but clustered in individuals when computing p-values and confidence intervals? If e.g. the 156 prescriptions to AD-patients are given to a small number of patients, the confidence intervals in table 6 would be wider than if they were given to 156 patients (10 prescriptions of the same drug to the same patient should count as one observation rather than as 10 independent observations).

Amount of drugs used: Is the amount of dispensed drugs proportional with number of prescriptions? The authors state that the prescriptions are dispensed for a maximum of 3-4 weeks, but is the duration the same for prescriptions with one drug and prescriptions with multipe drugs? Is there no information about number of tablets or DDDs (Defined Daily Doses; https://www.whocc.no/atc_ddd_index/)?

Prescribers: Since the authors’ recommendation is better intervention and they have access to the physicians’ identification numbers, could they tell whether prescribing multiple BZRAs is linked to specific prescribers or prescriber characteristica like age and sex?

Specific comments 

p2, line 59-61: Can the results from the two US studies (refs 7 and 9) – one of which is 15 years old – be generalized to China (or worldwide)?

p2, lines 71-72: “Recent retrospective studies have revealed lower levels of dependence in all age groups after long-term administration of BZRAs than those previously reported[26]”. It’s not so easy to find support for this statement in reference 26.

p3, line 116-117: “We processed the data at the level of the chemical therapeutic subgroup (N03AE, N05BA, N05CD, N05CF).” What does that mean? In table 3 the drugs are given at ATC level 5. Next sentence: “BZRAs offered by our hospital 117 include mainly BZDs, such as clonazepam, diazepam, lorazepam, alprazolam, and 118 eszopiclone, as well as Z-drugs, such as zopiclone and zolpidem.” Why “such as”?  In table 3 these are the only ones listed. Are there other ATC codes involved in prescriptions with more than one drug? Otherwise, I suggest to drop “such as”, add the ATC-codes from table 3 in parentheses after the drug names, and drop the sentence starting with “We processed the data...”.

p6, line 178: “BZRA consumption was higher among women with anxiety disorder than that among men”. This I interpret as “the proportion of women with anxiety disorder that use BZRAs is higher than the proportion of men with anxiety disorder that use BZRAs”, but these proportions we don’t know, only that the number of female anxiety patients that use BZRAs is higher than the number of male  anxiety patients that use BZRAs. I suggest a reformulation here.

p6, line 179: What does the p-value refer to, is it to each year separately? I suggest to include the percentages listed here in table 2.

p6, line 181: “Older age groups had a higher percentage of BZRA usage than that of younger age groups”. Again, this I interpret as “the proportion of old people that use BZRAs is higher than the proportion of old people that use BZRAs”, which we don’t know, only that there are more users over than under 45 years of age, which may be of limited interest. I suggest a reformulation.

p6, line 188-189: For age 45-64 the average was higher in 2020 (5.74) than in 2019 (5.59).

p6, line 196: Please replace “BZD” with “BZRA”.

p6, table 2: The numbers for the total in the two oldest age groups in 2018 do not equal the sum of men and women.

p7, line 205: Please replace “Figure.2 C” with “Figure 2 B”.

p7, line 209: I don’t understand “associated with increased prescriptions”. Does it mean “who obtained prescriptions including multiple BZRA drugs”?

p7, line 213: “The annual average number of prescriptions decreased dramatically over the 4-year period..” All of the decrease is from 2020 to 2021. Is there any explanation for this? Was it e.g. larger prescriptions in 2021 than in the previous years, as measured in number of tablets or DDDS? Was there a large number of new patients at the end of the year? Were there more patients with less serious anxiety in 2021? I assume that the complete year was covered until December 31?

p9, table 3: i) I think the title of the table is a bit misleading. The “propensity to be prescribed Clonazepam” I think is a measure related to the patient: the probability that a given patient will be prescribed Clonazepam, typically based on the patient’s background characteristica (covariates). However, the table just shows the distribution of BZRA prescriptions. ii) regarding “patients taking only one BZRA drug” in the title: If a patient first get a prescription of Clonazepam only and later a prescription of Diazepam only, is this patient and the two prescriptions not included? iii) do the p-values refer to sex-differences? Is clustering of prescriptions by individuals taken into account?

p10, lines 271ff: I suggest to move the methodological part of this description of coexisting conditions to the Methods section.

p11, table 6: Who are included in this analysis? The whole population, or only those with the conditions included as variables? Since the crude OR for women in this table is <1, whereas in tables 4 and 5 the proportion of women that have more than one drug is higher than the proportion of men, I suspect it is those with the conditions that are include. It would be informative if number of prescriptions and patients with 1 and more than 1 prescription per condition was added to the table (4 more columns). Why is not the “other” group included? Please replace “Sex” with “Women” under Variables.

p11, lines 283-288: If only prescriptions to patients with one of the specified comorbidities are included in the analyses,the results for age and sex are restricted to the comorbid population. Why isn’t the ‘other’ group included in the table?

p13, line 369. Regarding the marked decrease in average number of prescriptions in 2021, the authors refer to a Japanese study analysing data until March 2021. This study shows a gradual decrease in proportion of patients using BZRAs over the last years, but no decrease in mean average daily dose since 2015, which I guess is the most relevant result in this context. So I don’t think this article can be used for explaining the 2021 decline.  

p13, line 378: “While the concurrent use of multiple BZRAs may be unintentional, it often arises when patients have multiple prescribers and/or obtain a prescription that contains multiple BZRAs drugs.” Could that happen in this study? In which case the simultaneous use would underestimated?

p13, line 404: “In addition, when anxiety coexisted with depression, sleep disorders, and other chronic physical disorders, multiple BZRAs did not tend to be consumed simultaneously by the patients.” Is that compared to patients with the other specified comorbidities, or with the complete population?

Reviewer 2 Report

Thank you very much for the opportunity to review this manuscript.

I would like to make some contributions that will undoubtedly improve the quality of the manuscript:

-References 2 and 3 (and in general quite a few more) are too old for a subject that has been heavily researched in recent years. It is recommended that they be updated.

-Reference 3 and 4 are misused, as they do not deal with the economic impact of anxiety. Please check the adequacy of the references.

Example from text referenced as 3: This study assessed the role of two cognitive vulnerability factors, anxiety sensitivity and dysfunctional attitudes, in predicting the manifestation and onset of social anxiety disorder in relation to specific phobia and in relation to healthy controls. Females, aged 18-24 years

-Lines 49 and 50. The referenced text is from 2015; what is the most recent evidence in this regard? Please update the data

-Were only patients with an anxiety disorder considered, and what biases did the authors detect prior to conducting the study?

Round 2

Reviewer 2 Report

Dear authors, after reviewing the text again and checking that the necessary changes have been made, I think it is suitable for publication, except for some small details that need to be corrected. Congratulations
